# Diabetes and Second Neoplasia Impact on Prognosis in Pre-Fibrotic Primary Myelofibrosis

**DOI:** 10.3390/cancers14071799

**Published:** 2022-04-01

**Authors:** Daniele Cattaneo, Claudia Vener, Elena Maria Elli, Cristina Bucelli, Nicole Galli, Fabrizio Cavalca, Giuseppe Auteri, Donatella Vincelli, Bruno Martino, Umberto Gianelli, Francesca Palandri, Alessandra Iurlo

**Affiliations:** 1Hematology Division, Foundation IRCCS Ca’ Granda Ospedale Maggiore Policlinico, 20122 Milan, Italy; daniele.cattaneo@unimi.it (D.C.); cristina.bucelli@policlinico.mi.it (C.B.); nicole.galli@unimi.it (N.G.); 2Department of Oncology and Hemato-Oncology, University of Milan, 20122 Milan, Italy; claudia.vener@unimi.it; 3Hematology Division, San Gerardo Hospital, ASST Monza, 20900 Monza, Italy; elena.elli@libero.it (E.M.E.); cavalca.fabrizio@gmail.com (F.C.); 4Institute of Hematology “L. and A. Seràgnoli”, S. Orsola-Malpighi Hospital, 40138 Bologna, Italy; giuseppe.auteri2@unibo.it (G.A.); francesca.palandri@unibo.it (F.P.); 5Division of Hematology, Azienda Ospedaliera “Bianchi Melacrino Morelli”, 89133 Reggio Calabria, Italy; donatella.vincelli@gmail.com (D.V.); brunmartin54@gmail.com (B.M.); 6Department of Pathophysiology and Transplantation, University of Milan, 20122 Milan, Italy; umberto.gianelli@unimi.it; 7Division of Pathology, Foundation IRCCS Ca’ Granda Ospedale Maggiore Policlinico, 20122 Milan, Italy

**Keywords:** primary myelofibrosis, pre-fibrotic, prognosis, scoring system, IPSS, DIPSS, diabetes, second neoplasia

## Abstract

**Simple Summary:**

The 2016 WHO-revised classification of MPNs recognized pre-fibrotic PMF (pre-PMF) as a distinct clinical entity from both overt fibrotic PMF (overt PMF) and essential thrombocythemia (ET). In fact, while the initial presentation of pre-PMF is often an isolated thrombocytosis, thus mimicking ET, its course may be symptomatic in a non-negligible number of cases. Conversely, overt PMF patients are enriched in higher-risk categories, thus suggesting a greater propensity for disease progression than pre-PMF. Importantly, median survival is significantly reduced in overt PMF vs. pre-PMF, thereby reinforcing the appropriateness of making this distinction in clinical practice. Nevertheless, a specific prognostic model for pre-PMF is still lacking, except for thrombotic risk. The aim of the present study was therefore to identify covariates other than those commonly related to PMF, which can better define prognosis in pre-PMF patients in the real-world setting, thus resulting in more personalized and efficient therapeutic approaches.

**Abstract:**

The 2016 WHO classification recognized pre-fibrotic primary myelofibrosis (pre-PMF) as a distinct entity. Nevertheless, a prognostic model specific for pre-PMF is still lacking. Our aim was to identify the most relevant clinical, histological, and driver mutation information at diagnosis to evaluate outcomes in pre-PMF patients in the real-world setting. We firstly assessed the association between IPSS or DIPSS at diagnosis and response variables in 378 pre-PMF patients. A strict association was observed between IPSS and DIPSS and occurrence of death. Other analyzed endpoints were not associated with IPSS or DIPSS as thrombo-hemorrhagic events at diagnosis or during follow-up, or did not show a clinical plausibility, as transformation into acute leukemia or overt PMF. The only covariates which were significantly associated with death were diabetes and second neoplasia, and were therefore included in two different prognostic settings: the first based on IPSS at diagnosis [class 1 vs. 0, OR (95%CIs): 3.34 (1.85–6.04); class 2 vs. 0, OR (95%CIs): 12.55 (5.04–31.24)], diabetes [OR (95%CIs): 2.95 (1.41–6.18)], and second neoplasia [OR (95%CIs): 2.88 (1.63–5.07)]; the second with DIPSS at diagnosis [class 1 vs. 0, OR (95%CIs): 3.40 (1.89–6.10); class 2 vs. 0, OR (95%CIs): 25.65 (7.62–86.42)], diabetes [OR (95%CIs): 2.89 (1.37–6.09)], and second neoplasia [OR (95%CIs): 2.97 (1.69–5.24)]. In conclusion, our study underlines the importance of other additional risk factors, such as diabetes and second neoplasia, to be evaluated, together with IPSS and DIPSS, to better define prognosis in pre-PMF patients.

## 1. Introduction

Primary myelofibrosis (PMF), which belongs to the *BCR-ABL1*-negative myeloproliferative neoplasms (MPNs), is mainly characterized by atypical megakaryocyte proliferation, bone marrow fibrosis, extramedullary hematopoiesis, variable cytopenias, hepatosplenomegaly, constitutional symptoms, leukemic progression, and shortened survival [1].

Historically, PMF patients have been stratified using three prognostic scoring systems based on age, constitutional symptoms, and chemistry. These systems are the International Prognostic Scoring System (IPSS) [2], applicable only at diagnosis, the Dynamic International Prognostic Scoring System (DIPSS) [3], and the DIPSS-plus [4], which can be evaluated at any time.

However, factors other than clinical and laboratory features have important prognostic significance in PMF, such as bone marrow fibrosis (BMF), particularly when evaluated according to WHO criteria [5]. In a previous study by our group [6], we demonstrated that the BMF grade based on the EUMNET consensus [7] is significantly associated with overall survival in MPNs, and can more accurately discriminate between intermediate- and high-risk patients. These observations were then confirmed in a subsequent study in which an advanced BMF grade was significantly associated with clinical characteristics indicative of a more aggressive disease and additional prognostically adverse somatic mutations in the *ASXL1* and *EZH2* genes [8].

Alongside the negative prognostic impact of so-called high molecular risk (HMR) mutations, several studies have also recently identified host genetic variants and inflammatory/immune markers [9] as independent predictors for clinical evolution and reduced survival in PMF patients [10], including high-sensitivity C-reactive protein [11,12], sIL-2Rα [13], eNAMPT [14], and CXCR4 expression on circulating CD34+ cells [15] and CCL2 [16] and VEGF-A polymorphisms [17,18].

Nevertheless, as information on “other” mutations and aforementioned additional biological parameters are available in clinical practice only in a limited number of laboratories, we proposed to assess PMF prognosis at diagnosis by combining IPSS, BMF grade, and driver mutations profile, i.e., information derived from the “good clinical practice” management of PMF [19].

As a matter of fact, the 2016 WHO-revised classification of MPNs recognized pre-fibrotic PMF (pre-PMF) as a distinct clinical entity from both overt fibrotic PMF (overt PMF) and essential thrombocythemia (ET) [5,20]. Indeed, while the initial presentation of pre-PMF is often an isolated thrombocytosis, thereby mimicking ET, its course may be symptomatic in a non-negligible number of cases [21]. Pre-PMF patients typically have higher leukocytes and platelets, lower hemoglobin, higher lactate dehydrogenase (LDH), and, more frequent splenomegaly than ET [21]. Furthermore, pre-PMF may show a progressive clinical course with worsening of constitutional symptoms, increased BMF grade, and the appearance of high-risk cytogenetic or molecular abnormalities. However, it should be remembered that pre-PMF patients generally belong to lower prognostic risk categories at diagnosis. Conversely, overt PMF patients are enriched in higher-risk categories, not only at diagnosis but also during follow-up, thus suggesting a greater propensity for disease progression than with pre-PMF. Importantly, median survival is significantly reduced in overt PMF vs. pre-PMF (7.2 vs. 17.6 years) [21], thereby reinforcing the appropriateness of making this distinction in clinical practice.

In a recent study, Carobbio et al. specifically focused on disease progression rates in 372 pre-PMF patients to look for prognostic factors that predict transition through different phases of the disease, including overt PMF and acute myeloid leukemia (AML) [22]. Using a multistate model, the authors identified advanced age (>65 years) and leukocytosis (>15 × 10^9^/L) as predictors of death and AML, while risk factors for fibrotic progression included anemia and grade 1 BMF [22]. Interestingly, for models including mutational status, the presence of one or more HMR mutations was the only factor significantly associated with an increased risk of transition from pre- to overt PMF, AML, and death.

Nevertheless, although pre-PMF should be considered as being apart from both overt PMF and ET, there is still a lack of a prognostic model specific for this disease, except for thrombotic risk [23].

The aim of this study was therefore to identify covariates other than those commonly related to PMF, which can better define prognosis in pre-PMF patients in the real-world setting.

## 2. Materials and Methods

### 2.1. Patients

The present multicenter study included 378 consecutive pre-PMF patients who were diagnosed in four Italian hematological centers (Milan, Monza, Bologna, and Reggio Calabria) between November 1983 and December 2019, with a median follow-up of 7.9 years (range: 0.2–36.3 years). All cases were reviewed according to the WHO 2016 classification [5]. Inclusion criteria were as follows: demographic, clinical, hematological, and histological data available at the time of diagnosis; stored bone marrow; and at least one granulocyte DNA sample to assess the mutational status of *JAK2*, *CALR*, and *MPL* genes. Patients without mutations in the *JAK2*, *CALR*, and *MPL* genes were defined as “triple-negative”. Baseline clinical characteristics and outcome measures (death, fibrotic progression, leukemic evolution, and thrombo-hemorrhagic complications) were assessed. Clinical prognostic classification of patients at diagnosis was made according to IPSS and DIPSS [2,3]. For the corresponding risk categories, the intermediate-2 and high-risk classes were combined because the median survival of these patients is significantly shorter than that of both low- and intermediate-1 risk patients [2,3]. Consequently, the first two categories (intermediate-2 and high-risk) are commonly managed in the same way, including both drug and transplant strategies [24].

In all four hematological centers, patients were treated with antiplatelet and cytoreductive/targeted agents according to international guidelines, which were based on current recommendations [24]. Follow-up information was updated in December 2020.

### 2.2. Methods

#### 2.2.1. Pre-PMF Molecular Analyses

The *JAK2*V617F mutation was detected by allele-specific PCR according to the protocol of Baxter et al. [25] and confirmed by direct Sanger sequencing. Quantitative analysis of the allele burden of the *JAK2*V617F mutation was performed by RQ-PCR using JAK2 MutaQuant (Ipsogen Inc., New Haven, CT, USA). The cut-off used for defining a case as negative for *JAK2*V617F mutation was 0.5%.

*MPL* mutations, in particular W515L, W515K, W515A, S505N, and G509C, were tested by direct sequencing of exon 10. The primers used were as follows: MPL10F 5′ TAGCCTGGATCTCCTTGGTG 3′; MPL10R 5′ CCTGTTTACAGGCCTTCGGC 3′.

Mutations in exon 9 of the *CALR* gene were also assessed using a bidirectional sequencing approach, as previously described [26]. All sequencing analyses were performed on an ABI PRISM 310 Genetic Analyzer (Applied Biosystems, Warrington, UK) using the Big Dye Terminator v1.1 Cycle Sequencing Kit (Applied Biosystems).

#### 2.2.2. Bone Marrow Biopsy

Histologic confirmation of pre-PMF diagnosis, as defined by the 2016 WHO classification [5], was performed by experienced pathologists. Formalin-fixed, paraffin-embedded bone marrow biopsy samples obtained at diagnosis were available for all patients. Sections were stained with hematoxylin-eosin, Giemsa, and Gomori’s silver impregnation for evaluation of morphologic features and BMF.

#### 2.2.3. Statistical Analysis

Descriptive statistics were generated for all clinical characteristics and outcome measures as appropriate: for continuous variables, median, range; for categorical variables, frequency, percentage. Missing data were not implemented and were considered as lack of data.

First of all, to select the most relevant endpoint for subsequent analyses, univariable logistic regression analysis was performed to evaluate the association between IPSS and DIPSS at diagnosis [categorized into three prognostic classes: low (class 0); intermediate-1 (class 1); intermediate-2/high (class 2)] and response variables (occurrence of: death; thrombosis and hemorrhages at diagnosis or during follow-up; transformation into AML or overt PMF; composite outcome: occurrence of thrombosis or hemorrhage or transformation into AML or overt PMF), respectively (Table 2).

After selecting the most relevant endpoint, univariable (Table 3) and multivariable (Tables 4 and 5) logistic regression analyses were performed to evaluate the association between patients’ clinical and biological variables (covariates, including IPSS and DIPSS at diagnosis, respectively) and the selected response variable (endpoint, in the specific case occurrence of death). In the absence of significant differences in the duration of follow-up between living and deceased patients, a preliminary analysis was conducted to select the prognostic indexes to be considered in the future time-dependent survival analysis. An explanatory analysis was performed using logistic regression models to investigate the effects of covariates on patients’ status (alive/dead) [19]. The probability of death was modeled as a function of covariates, after removing those strongly associated with each other (χ^2^ test: *p* < 0.05).

Firstly, multivariable logistic regression analysis was performed including only covariates that achieved statistical significance with univariable logistic regression (*p* < 0.05) (Table 3). The probability of death was finally adapted with two different models: the first comprising the effects of IPSS at diagnosis, other patients’ covariates, and their interaction (Table 4); the second which included the effects of DIPSS at diagnosis, other patients’ covariates, and their interaction (Table 5).

Secondly, due to the large number of possible patients’ covariates (Table 3), and to detect which of them has the greatest influence on the probability of death, an automatic stepwise model selection approach was adopted, having set a significance level of 0.30 to insert a variable in the model and a significance level of 0.05 to maintain a variable in the model. We report SAS instructions to perform stepwise automatic selection (proc logistic). proc logistic data = …; model …(event = “…”) = …/selection = stepwise slentry = 0.3 slstay = 0.05; output out = pred p = phat lower = lcl upper = ucl predprob = (individual crossvalidate); ods output Association = Association; run; SAS statistical software (SAS version 9.4 of the SAS System, SAS Institute Inc., Cary, NC, USA).

Thirdly, a manual selection of the patients’ covariates was performed, including all available covariates at the start of the model analysis (Table 3), and removing the least significant one by one, considering a significance level of 0.05 to maintain a variable in the final model.

Odds Ratio (OR) with 95% confidence intervals (CIs) were calculated. All the analyses, summaries, and listings have been carried out, and all statistical models have been fitted using SAS statistical software (SAS version 9.4 of the SAS System, SAS Institute Inc., Cary, NC, USA).

## 3. Results

The complete dataset included 378 pre-PMF patients (median age: 64.9 years; range: 18.9–91.9 years; 46.0% male) and their key demographic, clinical, and laboratory features at diagnosis are shown in Table 1. Risk distribution according to the IPSS was low in 41.0% of patients, intermediate-1 in 47.1%, intermediate-2 in 8.5%, and high in the remaining 3.4%.

*JAK2*V617F mutation was detected in 256 cases (67.7%), *CALR* mutations in 79 (20.9%), and *MPL* in 10 (2.7%). Thirty-three patients (8.7%) were defined as “triple-negative”. *CALR* mutations were mutually exclusive with both *JAK2* and *MPL* mutations. Among the 79 *CALR*-mutated patients, 47 (59.5%) had type 1 and 22 (27.8%) had type 2 mutations; the remaining 10 cases (12.7%) carried other distinct variants.

According to the updated WHO 2016 classification [5], all cases had no BMF (MF-0, 37.6%) or only a slight increase in reticulin fibrosis (MF-1, 62.4%).

As reported in Table 2, we firstly assessed any possible association between IPSS or DIPSS at diagnosis, respectively, and response variables (occurrence of: death; thrombosis or hemorrhages at diagnosis or during follow-up; transformation into AML or overt PMF; composite outcome: occurrence of thrombosis or hemorrhage or transformation into AML or overt PMF). A strict association was observed between IPSS at diagnosis and occurrence of death [class 1 vs. 0, OR (95%CIs): 3.47 (1.97–6.12); class 2 vs. 0, OR (95%CIs): 10.56 (4.41–25.3)]; the same clear association was detected between DIPSS at diagnosis and occurrence of death [class 1 vs. 0, OR (95%CIs): 3.50 (2.00–6.12); class 2 vs. 0, OR (95%CIs): 20.66 (6.36–67.06)]. Other analyzed endpoints did not show a significant association with IPSS or DIPSS at diagnosis, respectively, such as thrombosis and hemorrhage at diagnosis or during follow-up, or did not show a clinical plausibility, such as transformation into AML or overt PMF, and composite outcome: [ORs (95%CIs) lower than 1] (Table 2). Accordingly, the occurrence of death was selected as the only endpoint for subsequent analyses.

Median follow-up duration for patients still alive at the last clinical evaluation was comparable to that of deceased patients: 8.7 years (range, 0.3–35.1; Q1, 5.6; Q3, 12.3) vs. 7.9 years (range, 0.9–36.3; quartile (Q) 1, 4.5; Q3, 10.6), respectively. Furthermore, since the follow-up times of both living and deceased patients showed a fairly normal distribution, comparison by means revealed no significant differences (mean, 9.4—standard deviation (SD), 6.0; mean, 8.5—SD, 5.8; in years, respectively; *t*-test: *p* = 0.173). Consequently, explorative analysis could be performed without the presence of censored patients.

Interestingly, fibrotic progression was reported in only 30 (7.9%) patients after a median follow-up from pre-PMF diagnosis of 7.2 years (range: 1.8–24.4 years). Leukemic evolution was instead documented in 25 (6.6%) cases after a median follow-up of 8.8 years (range: 1.6–18.2 years), with a previous diagnosis of overt PMF in only four cases.

Moreover, clear associations between the occurrence of death and patients’ clinical-biological variables (univariable analysis) were observed, in particular with: IPSS at diagnosis [class 1 vs. 0, OR (95%CIs): 3.47 (1.97–6.12); class 2 vs. 0, OR (95%CIs): 10.56 (4.41–25.30)], DIPSS at diagnosis [class 1 vs. 0, OR (95%CIs): 3.50 (2.00–6.12); class 2 vs. 0, OR (95%CIs): 20.66 (6.36–67.06)] (Table 2 and Table 3), presence of diabetes [OR (95%CIs): 2.99 (1.53–5.87)], hypertension [OR (95%CIs): 1.85 (1.14–2.99)], positive thrombophilia screening [OR (95%CIs): 2.04 (1.02–4.11)], second neoplasia [OR (95%CIs): 2.59 (1.54–4.34)] and transfusions requirement [OR (95%CIs): 9.04 (4.60–17.75)] (Table 3).

Furthermore, we estimated the association between covariates listed in Table 3 and the occurrence of death (multivariable analysis). We excluded transfusions covariate from subsequent analysis because it is strongly associated with both IPSS and DIPSS score at diagnosis (χ^2^ test: *p* < 0.0001); the same was true for hypertension covariate (χ^2^ test: *p* = 0.0036 for IPSS and *p* = 0.0033 for DIPSS score at diagnosis). We also excluded the thrombophilia screening covariate because it is strongly associated with the onset of diabetes (χ^2^ test: *p* = 0.0002) and DIPSS score at diagnosis (χ^2^ test: *p* = 0.0388). Finally, two different prognostic settings were defined: the first model included IPSS score at diagnosis [class 1 vs. 0, OR (95%CIs): 3.34 (1.85–6.04); class 2 vs. 0, OR (95%CIs): 12.55 (5.04–31.24)], presence of diabetes [OR (95%CIs): 2.95 (1.41–6.18)], and second neoplasia [OR (95%CIs): 2.88 (1.63–5.07)] (Table 4) [Akaike information criterion (AIC) 358.862 for IPSS (univariable model); AIC 340.160 for IPSS, diabetes and second neoplasia (multivariable model)]; the second model included DIPSS score at diagnosis [class 1 vs. 0, OR (95%CIs): 3.40 (1.89–6.10); class 2 vs. 0, OR (95%CIs): 25.65 (7.62–86.42)], presence of diabetes [OR (95%CIs): 2.89 (1.37–6.09)], and second neoplasia [OR (95%CIs): 2.97 (1.69–5.24)] (Table 5) [AIC 354.421 for DIPSS (univariable model); AIC 335.466 for DIPSS, diabetes and second neoplasia (multivariable model)].

## 4. Discussion

The revised 2016 WHO classification of myeloid malignancies dictated distinct criteria for pre- and overt PMF [5], which are mainly based on bone marrow morphology and the degree of fibrosis with BMF grade of 0 and 1 included in the pre-PMF category. In addition, peripheral blood leukoerythroblastosis is a minor diagnostic criterion for overt PMF, whereas anemia, leukocytosis, increased LDH, and palpable splenomegaly may be present in both diseases [5].

The existence of pre-PMF as a separate entity, and its differentiation from strictly WHO-defined ET, has been debated for several years [27], sometimes with conflicting results [28,29]. Low interobserver agreement in the application of WHO-based histopathological criteria for pre-PMF has been questioned by some experts [30], while others have clearly delineated their reproducibility and the clinical relevance of adopting the diagnostic concept of pre-PMF [31]. In the largest multicenter study reported in the literature so far, 1104 ET patients underwent a central re-review of their diagnostic biopsies. The diagnosis of ET was then confirmed in 891 (81%) patients, while 180 (16%) were reclassified as pre-PMF, with important prognostic implications. Indeed, when compared with ET, the 10-year (76% vs. 89%) and 15-year survival rates (59% vs. 80%), leukemic transformation rates at 10 (5.8% vs. 0.7%) and 15 years (11.7% vs. 2.1%), as well as the progression rates to overt PMF at 10 (12.3% vs. 0.8%) and 15 years (16.9% vs. 9.3%) were all significantly worse in pre-PMF patients. Multivariable analysis confirmed these results and identified age > 60 years, leukocyte count > 11 × 10^9^/L, anemia, and history of thrombosis as additional risk factors for survival, further underscoring the importance of differentiating pre-PMF from ET, particularly with regards to pre-PMF patients with absent fibrosis [32].

In this context, the appropriateness of a reappraisal of IPSS has been questioned; in fact, it was originally developed using PMF patients [2] that differed at least in part from the two categories of pre- and overt PMF currently identified by the 2016 revised WHO criteria [5]. In this regard, Guglielmelli et al. found that, although IPSS overall predicted survival, it largely failed to accurately distinguish between intermediate-1 and intermediate-2, and intermediate-2 and high-risk patients, respectively, in pre- and overt PMF, as well as in the individual groups according to the degree of fibrosis [21]. These observations may have importance in the settings of the decision-making process for stem cell transplantation, which is currently indicated in intermediate-2/high risk, as well as in selected intermediate-1 risk PMF patients [33], with a non-negligible percentage of subjects being inappropriately exposed to such a risky procedure.

Collectively, these findings should promote efforts to critically reassess current prognostic scores and ultimately develop separate risk scores for pre- and overt PMF that include the most relevant clinical, histological, molecular, and cytogenetic variables.

The present multicenter study identifies two clinical variables to integrate prognostic information from the two well-known prognostic models for PMF, i.e., IPSS and DIPSS, and refine the prognosis in pre-PMF patients.

In this specific context, even though the degree of BMF (MF-0/1 vs. MF-2/3) has already been shown to play a crucial role in better defining PMF prognosis [19], neither BMF grade (MF-0 vs. MF-1) nor driver mutation status seems to exert any significant impact on the outcome. The first of these two observations further confirms the importance of a correct diagnosis of pre-PMF vs. overt PMF because, from a histological point of view, the main prognostic risk factor is represented by a BMF degree ≥ 2, as it has already been reported in the two most recent prognostic models developed for PMF, namely MIPSS70 and MIPSS70+ version 2.0 [34,35].

However, it should be underlined further that a prognostic model specific for pre-PMF is still lacking.

Interestingly, in our series, the strongest association with outcome was documented for the two models that included a common cardiovascular risk factor, like diabetes, and the occurrence of secondary malignancies, either hematological or not.

A similar observation of the impact of comorbidities on prognosis and outcome in cancer patients has already been made for other hematological malignancies. In 2011, Naqvi et al. conducted a retrospective cohort study of 600 consecutive patients with myelodysplastic syndromes (MDS), applying the Adult Comorbidity Evaluation-27 (ACE-27) scale to evaluate comorbidities [36]. Considering patients with no, mild, moderate, or severe comorbidities, median survival progressively decreased from 31.8 to 16.8, 15.2, and 9.7 months, respectively (*p* < 0.001), regardless of age and IPSS risk group; accordingly, a thorough assessment of comorbidity severity can help predict survival in MDS patients.

Similarly, in 2014, Newberry et al. evaluated the frequency and severity of comorbidities in 349 consecutive PMF patients [37]. As expected, comorbidities had a significant negative impact on survival (*p* < 0.001), with subjects suffering from severe comorbidities having double the risk of death compared to those without comorbidities. However, this study only considered patients who were diagnosed between 2000 and 2008, i.e., using the 2008 WHO criteria with no distinction between pre- and overt PMF.

Being aware of the limitation of the present study, represented mainly by its retrospective design, it can be hypothesized that disease progression (whether overt PMF or AML) represented the final cause of death only in a minority of cases (28/107, 26.2%), while a leading role is played by other events, including thrombo-hemorrhagic complications (13/107, 12.1%), or due to other neoplasia or related treatments (21/107, 19.6%), thus confirming how pre-PMF may represent a chronic disease with possible multiorgan involvement.

In such a context, it should still be remembered that pre-PMF patients have approximately a two times greater risk of cardiovascular events, including major thromboses and hemorrhages, compared to the reference age-matched population [23]. Indeed, in a recent paper [23], Guglielmelli et al. demonstrated that the risk of total thromboses in pre-PMF can be accurately predicted by the IPSET score, originally developed for ET [38], corresponding to 0.67, 2.05, and 2.95% patients/year in the low-, intermediate-, and high-risk categories, thus representing the basis for individualized management aimed at reducing the increased risk of major cardiovascular events in this specific subgroup of PMF patients.

Furthermore, in addition to an inherent risk of thrombo-hemorrhagic events, recent studies have consistently reported that MPNs are also prone to developing second cancers (SC) [39], and the latter can have a negative impact on MPN outcome; in particular, in a recent large international study including 1881 cases [40], patients were grouped into two prognostic classes based on the five-year relative survival from cancer diagnosis, with a “poor prognosis” SC group including cancers in the stomach, esophagus, liver, pancreas, lung, ovary, head-and-neck, nervous system, osteosarcomas, multiple myeloma, aggressive lymphoma, and acute leukemia [41]. In addition, MPN patients with SC have already shown to be exposed to an increased risk of arterial thromboses; indeed, thrombotic events after MPN and before SC were higher in cases than in controls without a history of SC (11.6% vs. 8.1%; *p* = 0.013), due to a higher rate of arterial thromboses (6.2% vs. 3.7%; *p* = 0.015) [42].

Another limitation of this study is the lack of a validation cohort. Due to the low representativeness of some explanatory variables, not completely available to all patients, it was not possible to create two different analysis groups (the setting cohort and the validation cohort).

## 5. Conclusions

This comprehensive approach allows clinicians to better assess pre-PMF prognosis, thus identifying high-risk patients with an unfavorable outcome, using a simple tool which can be easily applied worldwide since it only requires information from the two historical well-known prognostic models for PMF, namely IPSS and DIPSS at diagnosis, respectively, together with diabetes and second neoplasia.

Importantly, we also reinforce the need for careful differentiation between pre- and overt PMF, not only for a correct prognostic stratification, but also to allow physicians to apply all the preventive strategies to improve cardiovascular risk factors, including diabetes, with the goal of avoiding thrombo-hemorrhagic events. In addition, a careful surveillance of the possible co-existence of secondary malignancies in these patients from diagnosis should be indicated.

## Figures and Tables

**Table 1 cancers-14-01799-t001:** Baseline clinical and laboratory features of 378 pre-PMF patients.

Clinical-Laboratory Features	Patients (n = 378)
Male/Female	174/204
Age (years), median (range)	64.9 (18.9–91.9)
Hb (g/dL), median (range)	13.7 (6.8–19.7)
Hct (%), median (range)	41.6 (20.3–60.3)
WBC count (×10^9^/L), median (range)	9.1 (2.1–42.6)
PLT count (×10^9^/L), median (range)	687 (51–2513)
Peripheral blood blasts ≥ 1%, n. (%)	19 (5.0)
LDH (IU/L), median (range)	441.5 (119–2960)
Serum erythropoietin (IU/L), median (range)	6.6 (0.04–1002)
Circulating CD34+ cells (/µL), median (range)	5.0 (0–330)
Constitutional symptoms, n. (%)	38 (10.1)
Palpable splenomegaly, n. (%)	140 (37.0)
IPSS, n. (%)	
Low risk	155 (41.0)
Intermediate-1 risk	178 (47.1)
Intermediate-2 risk	32 (8.5)
High risk	13 (3.4)
DIPSS, n. (%)	
Low risk	155 (41.0)
Intermediate-1 risk	196 (51.8)
Intermediate-2 risk	26 (6.9)
High risk	1 (0.3)
Cytogenetic abnormalities, n. (%)	44 (11.6)
*JAK2*V617F, n. (%)	256 (67.7)
*JAK2* allele burden (%), median (range)	29.0 (1.4–99.1)
*CALR* mutations, n. (%)	79 (20.9)
Type 1 mutation, n. (%)	47 (12.4)
Type 2 mutation, n. (%)	22 (5.8)
Other mutations, n. (%)	10 (2.7)
*MPL* mutations, n. (%)	10 (2.7)
Triple-negative, n. (%)	33 (8.7)
Reticulin fibrosis grade, n. (%)	
MF-0	142 (37.6)
MF-1	236 (62.4)
Comorbidities, n. (%)	
Diabetes	50 (13.2)
SC	102 (27.0)
Hematologic malignancies	8 (2.1)
Non-hematologic malignancies	94 (24.9)
Thrombotic events, n. (%)	
At diagnosis	23 (6.1)
During follow-up	76 (20.1)
Hemorrhagic events, n. (%)	
At diagnosis	/
During follow-up	40 (10.6)
Disease progression, n. (%)	
overt PMF	30 (7.9)
AML	25 (6.6)
Deceased, n. (%)	107 (28.3)
Disease progression (including AML)	28 (7.4)
Thrombo-hemorrhagic events	13 (3.5)
Infectious complications	14 (3.7)
Other unrelated causes (including SC)	25 (6.6)
Unknown	27 (7.1)
Lost to follow-up, n. (%)	78 (20.6)
Cytoreductive/targeted therapy, n. (%)	304 (80.4)
Hydroxyurea	292 (77.3)
Ruxolitinib	47 (12.4)
Antiplatelet therapy, n. (%)	309 (81.8)

Abbreviations: Hb: hemoglobin; Hct: hematocrit; WBC: white blood cells; PLT: platelets; LDH: lactate dehydrogenase; IPSS: International Prognostic Scoring System; DIPSS: Dynamic International Prognostic Scoring System; SC: second cancer; AML: acute myeloid leukemia.

**Table 2 cancers-14-01799-t002:** Univariable analysis considering the association between IPSS or DIPSS at diagnosis and all possible outcomes.

Covariate	All Patients (N)	Response Variable	Class Comparison	OR (95%CI)	*p* Value *	*p* Value *,$
IPSS at diagnosis	300	Death	1 vs. 0	3.47 (1.97–6.12)	<0.000 *	0.813
IPSS at diagnosis	300	Death	2 vs. 0	10.56 (4.41–25.3)		<0.000 *
DIPSS at diagnosis	300	Death	1 vs. 0	3.5 (2.00–6.12)	<0.000 *	0.442
DIPSS at diagnosis	300	Death	2 vs. 0	20.66 (6.36–67.06)		<0.000 *
IPSS at diagnosis	378	Thrombosis ^	1 vs. 0	0.71 (0.43–1.17)	0.302	0.675
IPSS at diagnosis	378	Thrombosis ^	2 vs. 0	0.63 (0.28–1.42)		0.464
DIPSS at diagnosis	378	Thrombosis ^	1 vs. 0	0.71 (0.44–1.15)	0.291	0.834
DIPSS at diagnosis	378	Thrombosis ^	2 vs. 0	0.57 (0.20–1.61)		0.452
IPSS at diagnosis	378	Hemorrhage	1 vs. 0	1.31 (0.62–2.74)	0.584	0.985
IPSS at diagnosis	378	Hemorrhage	2 vs. 0	1.68 (0.60–4.71)		0.419
DIPSS at diagnosis	378	Hemorrhage	1 vs. 0	1.31 (0.63–2.71)	0.541	0.896
DIPSS at diagnosis	378	Hemorrhage	2 vs. 0	1.90 (0.57–6.33)		0.377
IPSS at diagnosis	378	Transformation #	1 vs. 0	0.33 (0.16–0.66)	0.007 *	0.008 *
IPSS at diagnosis	378	Transformation #	2 vs. 0	0.77 (0.31–1.89)		0.514
DIPSS at diagnosis	378	Transformation #	1 vs. 0	0.35 (0.18–0.67)	0.006 *	0.006 *
DIPSS at diagnosis	378	Transformation #	2 vs. 0	0.95 (0.33–2.71)		0.362
IPSS at diagnosis	378	Composite outcome °	1 vs. 0	0.60 (0.39–0.94)	0.070	0.212
IPSS at diagnosis	378	Composite outcome °	2 vs. 0	0.65 (0.33–1.30)		0.603
DIPSS at diagnosis	378	Composite outcome °	1 vs. 0	0.60 (0.39–0.93)	0.068	0.211
DIPSS at diagnosis	378	Composite outcome °	2 vs. 0	0.70 (0.30–1.62)		0.795

R, Odds Ratio; 95%CI, 95% confidence interval. (^) Thrombosis: at diagnosis or during follow-up. (#) Transformation into: AML or overt PMF. (°) Composite outcome: occurrence of thrombosis or hemorrhage or transformation into AML or overt PMF. Due to the low number of events, thrombosis and hemorrhage occurrence was considered cumulatively (at diagnosis plus during disease course); ORs (95%CIs) statistically significant in bold; (*) *p* < 0.05; ($) by class, according with SAS statistical software parameters.

**Table 3 cancers-14-01799-t003:** Univariable analysis considering the association between covariates and patients’ outcome (occurrence of death, only).

Covariate	All Patients (N)	Class Comparison °	OR (95%CI)	*p* Value *	*p* Value *,$
Gender	300	M vs. F	1.39 (0.87–2.23)	0.174	-
IPSS at diagnosis °	300	1 vs. 0	3.47 (1.97–6.12)	<0.000 *	0.813
IPSS at diagnosis °	300	2 vs. 0	10.56 (4.41–25.30)		<0.000 *
DIPSS at diagnosis °	300	1 vs. 0	3.50 (2.00–6.12)	<0.000 *	0.442
DIPSS at diagnosis °	300	2 vs. 0	20.66 (6.36–67.06)		<0.000 *
Bone marrow fibrosis grade &	300	1 vs. 0	0.93 (0.57–1.54)	0.788	-
Driver mutations profile $1	300	1–2 vs. 0	1 vs. 0: 1.96 (0.89–4.33)	0.242	0.172
			2 vs. 0: 1.61 (0.52–4.99)		0.769
Driver mutations profile $2	300	1–5 vs. 0	-	0.551	-
Cytogenetic at diagnosis	252	1 vs. 0	1.47 (0.72–3.01)	0.287	-
Cytogenetic risk at diagnosis &	252	1 vs. 0	1.00 (0.33–3.09)	0.333	0.414
Cytogenetic risk at diagnosis &	252	2 vs. 0	3.01 (0.70–12.90)		0.161
Splenomegaly	297	1 vs. 0	1.49 (0.91–2.44)	0.113	-
Erythropoietin #	106	1 vs. 0	1.63 (0.70–3.78)	0.256	-
Transfusions	292	1 vs. 0	9.04 (4.60–17.75)	<0.000 *	-
Smoke ^	291	1 vs. 0	1.75 (0.93–3.27)	0.219	0.144
Smoke ^	291	2 vs. 0	1.14 (0.56–2.34)		0.696
Hypertension	298	1 vs. 0	1.85 (1.14–2.99)	0.012 *	-
Diabetes	300	1 vs. 0	2.99 (1.53–5.87)	0.001 *	-
Dyslipidemia	300	1 vs. 0	1.19 (0.71–1.98)	0.505	-
Family thrombosis history	178	1 vs. 0	1.09 (0.55–2.19)	0.801	-
Personal thrombosis history	294	1 vs. 0	1.32 (0.74–2.34)	0.349	-
Thrombophilia screening	167	1 vs. 0	2.04 (1.02–4.11)	0.045 *	-
Second neoplasia	300	1 vs. 0	2.59 (1.54–4.34)	0.000 *	-

OR, Odds Ratio; 95%CI, 95% confidence interval; M, male; F, female. (°) Reference scoring IPSS and DIPSS classes: low (L, class 0): 0-IPSS (or DIPSS) risk score value; intermediate (I, class 1): 1-IPSS (or DIPSS) risk score value; high risk (H, class 2): 2-, 3-IPSS (or DIPSS) risk score values; reference: L (class 0). (&) Bone marrow fibrosis grade: MF-0, MF-1; reference: MF-0. ($) Driver mutations profile: 0–2 classes (type 1 *CALR*, class 0; other *CALR* mutations/*JAK2*V617F/*MPL* mutations, class 1; triple-negative, class 2) ($1), 0–5 classes (type 1 *CALR*, class 0; type 2 *CALR*, class 1; triple-negative, class 2; other *CALR* mutations, class 3; *JAK2*V617F, class 4; *MPL* mutations, class 5) ($2) (from favourable to progressive increasing risk class); reference: negative (0). (&) Cytogenetic risk at diagnosis: 0, favourable; 1, unfavourable; 2, very high risk. (#) Erythropoietin: 0, normal range values: 4.3–29.0 IU/L; 1, abnormal values: <4.3 IU/L or >29.0 IU/L. (^) Smoke: 0, never; 1, previous; 2, now. ORs (95%CIs) statistically significant in bold; (*) *p* < 0.05; ($) by class, according with SAS statistical software parameters.

**Table 4 cancers-14-01799-t004:** Multivariable analysis considering the association between covariates and patients’ outcome (occurrence of death). Number of patients analysed: 300.

Covariate	Class Comparison	OR (95%CI)	*p* Value *	*p* Value *,$
IPSS at diagnosis °	1 vs. 0	3.34 (1.85–6.04)	<0.000 *	0.841
IPSS at diagnosis °	2 vs. 0	12.55 (5.04–31.24)		<0.000 *
Diabetes *	1 vs. 0	2.95 (1.41–6.18)	0.004 *	-
Second neoplasia *	1 vs. 0	2.88 (1.63–5.07)	0.000 *	-

AIC, Akaike information criterion; OR, Odds Ratio; 95%CI, 95% confidence interval. (°) Reference scoring IPSS classes: low (L, class 0): 0-IPSS risk score value; intermediate (I, class 1): 1-IPSS risk score value; high risk (H, class 2): 2-, 3-IPSS risk score values; reference: L (class 0). (*) Reference: absence. Interaction tests between covariates: not statistically significant. ORs (95%CIs) statistically significant in bold; (*) *p* < 0.05; ($) by class, according with SAS statistical software parameters. AIC: 358.862 for IPSS (univariable model); AIC: 340.160 for IPSS, diabetes and second neoplasia (multivariable model).

**Table 5 cancers-14-01799-t005:** Multivariable analysis considering the association between covariates and patients’ outcome (occurrence of death). Number of patients analysed: 300.

Covariate	Class Comparison	OR (95%CI)	*p* Value *	*p* Value *,$
DIPSS at diagnosis °	1 vs. 0	3.40 (1.89–6.10)	<0.000 *	0.257
DIPSS at diagnosis °	2 vs. 0	25.65 (7.62–86.42)		<0.000 *
Diabetes *	1 vs. 0	2.89 (1.37–6.09)	0.005 *	-
Second neoplasia *	1 vs. 0	2.97 (1.69–5.24)	0.000 *	-

AIC, Akaike information criterion; OR, Odds Ratio; 95%CI, 95% confidence interval. (°) Reference scoring DIPSS classes: low (L, class 0): 0-DIPSS risk score value; intermediate (I, class 1): 1-DIPSS risk score value; high risk (H, class 2): 2-, 3-DIPSS risk score values; reference: L (class 0). (*) Reference: absence. Interaction tests between covariates: not statistically significant. ORs (95%CIs) statistically significant in bold; (*) *p* < 0.05; ($) by class, according with SAS statistical software parameters. AIC: 354.421 for DIPSS (univariable model); AIC: 335.466 for DIPSS, diabetes and second neoplasia (multivariable model).

## Data Availability

The original contributions presented in the study are included in the article. Further inquiries can be directed to the corresponding author.

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
