# Peer review of "Diabetes and Second Neoplasia Impact on Prognosis in Pre-Fibrotic Primary Myelofibrosis"

_cancers, 2022, doi:10.3390/cancers14071799_

Round 1
Reviewer 1 Report
The manuscript by Cattaneo et al. highlights the clinical need for a prognostic model specific for prePMF. Pre-fibrotic PMF is considered a distinct pathological entity from both ET and overtPMF, with different clinical and prognostic implications. Although prePMF (compared to overtPMF) belongs to the lower prognostic risk category, it is typified by heterogeneous phenotypes and progressive clinical course that make patient risk stratification extremely challenging. In the era of molecular and genetic risk stratification, the Authors propose two novel scoring systems which can be easily applied worldwide to improve prognostic prediction of prePMF, combining the well-known and widely used IPSS and DIPSS scoring systems with the presence/absence of diabetes and second cancers. However the idea is not entirely new, since the evaluation of comorbidities as part of prognostic risk assessment has been demonstrated to improve risk stratification in patients with myelodysplastic syndromes (Naqvi et al., 2011) and comorbidities such as diabetes and secondary cancers, have a significant impact on the outcome of PMF patients (Newberry et al., 2014).
Moreover, the two suggested models resulted useful to predict death occurrence, but no association has been found between IPSS, DIPSS and transformation into AML, overtPMF or composite outcome, respectively. However, the prognostic performance of these two models in prediction patient survival was not evaluated as compared to IPSS and DIPSS alone, and a validation cohort is lacking.
Furthermore, it is not clear how the two suggested model could be applied to clinical setting, which is the weight of each predictor? Is it possible to extrapolate from the two models a weighted combined score to stratify prePMF patients?
Table 1, reporting baseline clinical and laboratory features of prePMF patients, is not complete. Data about thrombotic and hemorrhagic events at the time of the diagnosis and during follow-up, the number of patients with diabetes and second cancers, as well as causes of death are lacking and should be added in table 1, since they are considered for statistical analysis and then discussed in the results section.
Moreover, did the Authors find any associations between diabetes /second cancers and age, gender, IPSS and DIPSS categories? May type and severity of both diabetes and second cancers have a prognostic impact on prePMF survival? Please discuss.
The authors did not find any significant associations between IPSS and DIPSS at the diagnosis and thrombotic events or hemorrhage. It is not clear whether thrombotic and hemorrhagic events are considered at the diagnosis or during disease course, since most of patients were on therapy with hydroxyurea (78.6% of patients), antiplatelet drugs (83.1%) and ruxolitinib (12.7%), and these therapeutic approaches impact on thrombotic and hemorrhagic risk. Therefore, therapy could represent a bias for the statistical analysis, please discuss this point.
Recently, several studies have identified high-molecular risk mutations, host genetic variants and inflammatory markers as independent predictive factors for clinical evolution and reduced survival in PMF (Tefferri, 2020 doi.org/10.1002/ajh.26050; Campanelli et al., 2021 10.3390/cancers13215324; Villani et al., 2021 10.3390/genes12081271, Masselli et al., 2021 10.3390/cancers13112552; Masselli et al., 2020 10.3390/cancers13112552), do you think that some of these factors could also be used to improve prognosis in prePMF?
Moreover, table 4a and table 4b show the results of multivariate analysis and should be moved from methods section to results section. P values should be showed in all tables, and the stepwise selection model adopted for the choice of the best predictors is not clear, please clarify.
Author Response
The manuscript by Cattaneo et al. highlights the clinical need for a prognostic model specific for pre-PMF. Pre-fibrotic PMF is considered a distinct pathological entity from both ET and overt PMF, with different clinical and prognostic implications. Although pre-PMF (compared to overt PMF) belongs to the lower prognostic risk category, it is typified by heterogeneous phenotypes and progressive clinical course that make patient risk stratification extremely challenging. In the era of molecular and genetic risk stratification, the Authors propose two novel scoring systems which can be easily applied worldwide to improve prognostic prediction of pre-PMF, combining the well-known and widely used IPSS and DIPSS scoring systems with the presence/absence of diabetes and second cancers. However, the idea is not entirely new, since the evaluation of comorbidities as part of prognostic risk assessment has been demonstrated to improve risk stratification in patients with myelodysplastic syndromes (Naqvi et al., 2011) and comorbidities such as diabetes and secondary cancers, have a significant impact on the outcome of PMF patients (Newberry et al., 2014).
We thank the reviewer for his/her comments. As required, we have added and commented on the above reported studies in the “Discussion” section of our manuscript.
Moreover, the two suggested models resulted useful to predict death occurrence, but no association has been found between IPSS, DIPSS and transformation into AML, overt PMF or composite outcome, respectively. However, the prognostic performance of these two models in prediction patient survival was not evaluated as compared to IPSS and DIPSS alone, and a validation cohort is lacking.
The prognostic values of univariable models with IPSS or DIPSS alone have now been added in Table 2 and Table 3. As suggested by the reviewer, we have also added the AIC values to facilitate the comparison between the univariable and multivariable models (IPSS or DIPSS alone vs. IPSS or DIPSS together with diabetes and second neoplasia). In particular: AIC 358.862 for IPSS (univariable model); AIC 340.160 for IPSS, diabetes and second neoplasia (multivariable model); AIC 354.421 for DIPSS (univariable model); AIC 335.466 for DIPSS, diabetes and second neoplasia (multivariable model) [Modified in lines 246-247; lines 250-251; Tables 4a and 4b, legend]. Unfortunately, validation cohort is missing; due to the lack of data from some covariates, we used the cohort of all patients to define the models. We were unable to split the cohort into two different patients’ groups, firstly to define the model and secondly to validate our results, because the sample size was reduced for some explanatory variables. We have added this limitation in the revised version of our manuscript [Modified in lines 341-344].
Furthermore, it is not clear how the two suggested model could be applied to clinical setting, which is the weight of each predictor? Is it possible to extrapolate from the two models a weighted combined score to stratify pre-PMF patients?
We performed only a preliminary analysis to select covariates (other than IPSS and DIPSS) that may be associated with relevant patients’ outcomes. Unfortunately, only death showed statistical significance. We analyzed survival as a dichotomous outcome (alive vs. dead); we did not perform a time-dependent event survival analysis. This was possible because follow-up survival times for living and dead patients were comparable [Modified in lines 169-173; 218-224; lines 352-354]. However, it was not possible to run the Cox proportional hazard model (or similar) in the first place, starting from a huge number of covariates, different outcomes, a cohort with a limited number of patients and low representativeness of some covariates (some of which not fully available to all patients). We have added this limitation in the revised version of our manuscript [Modified in lines 341-344].
Table 1, reporting baseline clinical and laboratory features of pre-PMF patients, is not complete. Data about thrombotic and hemorrhagic events at the time of the diagnosis and during follow-up, the number of patients with diabetes and second cancers, as well as causes of death are lacking and should be added in table 1, since they are considered for statistical analysis and then discussed in the results section.
We have now added all the information he/she required in Table 1.
Moreover, did the Authors find any associations between diabetes /second cancers and age, gender, IPSS and DIPSS categories? May type and severity of both diabetes and second cancers have a prognostic impact on pre-PMF survival? Please discuss.
“Gender” was not associated with outcome (death, univariable analysis, Table 3). “Age” was already included in IPSS and DIPSS scoring systems, so it was not included in the final model as a separate covariate. For the other covariates, before being introduced in the multivariable model, the association between them has been excluded (multicollinearity). Multivariable analysis made it possible to adjust for other covariates at the same time. Unfortunately, we had no information on the severity of diabetes. As for the type of second neoplasms, they are almost all non-hematologic malignancies [Modified in Table 1].
The authors did not find any significant associations between IPSS and DIPSS at the diagnosis and thrombotic events or hemorrhage. It is not clear whether thrombotic and hemorrhagic events are considered at the diagnosis or during disease course, since most of patients were on therapy with hydroxyurea (78.6% of patients), antiplatelet drugs (83.1%) and ruxolitinib (12.7%), and these therapeutic approaches impact on thrombotic and hemorrhagic risk. Therefore, therapy could represent a bias for the statistical analysis, please discuss this point.
We have added in thrombotic and hemorrhagic events by occurrence at diagnosis and during follow-up to Table 1. Due to the low number of events, we considered them all together (thrombotic at diagnosis plus during follow-up and the same for hemorrhagic events). We know the limits of the interference between thrombotic and hemorrhagic events and therapy, but we have performed only preliminary analyses to select covariates and outcomes for future time-dependent survival analyses. As we found that the only significant outcome was death, other clinical outcomes (such as thrombotic and hemorrhagic events) will no longer be considered [Modified in Table 2, legend].
Recently, several studies have identified high-molecular risk mutations, host genetic variants and inflammatory markers as independent predictive factors for clinical evolution and reduced survival in PMF (Tefferri, 2020 doi.org/10.1002/ajh.26050; Campanelli et al., 2021 10.3390/cancers13215324; Villani et al., 2021 10.3390/genes12081271, Masselli et al., 2021 10.3390/cancers13112552; Masselli et al., 2020 10.3390/cancers13112552), do you think that some of these factors could also be used to improve prognosis in pre-PMF?
As required, we have added and commented on the above reported studies in the “Introduction” section of our manuscript.
Moreover, table 4a and table 4b show the results of multivariate analysis and should be moved from methods section to results section. P values should be showed in all tables, and the stepwise selection model adopted for the choice of the best predictors is not clear, please clarify.
We thank the reviewer for this suggestion, but we believe that reporting in Materials and Methods Section only the name (not the results) of the related tables can facilitate the reading of the methodological section. However, we can delete them as requested by the reviewer.
We have added all p values as required.
Here are the SAS instructions to perform stepwise automatic selection (proc logistic) [Modified in lines 184-185].
proc logistic data= …;
model …(event="…") = … /selection=stepwise slentry=0.3 slstay=0.05;
output out=pred p=phat lower=lcl upper=ucl predprob=(individual crossvalidate);
ods output Association=Association;
run;
SAS statistical software (SAS version 9.4 of the SAS System, SAS Institute Inc., Cary, NC, USA).
Reviewer 2 Report
In this study, authors evaluated covariates other than those commonly related to PMF which can better define prognosis in pre-PMF patients in the real-world setting. Their findings underlining the importance of other additional risk factors, such as diabetes and second neoplasia together with IPSS and DIPSS, can fill the gap to better define prognosis in pre-PMF patients. However, there are major concerns that should be addressed by the authors.
- In this study, an association was observed between IPSS and DIPSS and occurrence of death, whereas other analyzed endpoints were not associated with IPSS or DIPSS. Causes of death need to be summarized. Did events of death include only disease-related death or death which was both related and not related to disease?
- For second neoplasia, if possible, the information what kind of neoplasia was occurred may be useful. Did you include secondary hematologic malignancies as second neoplasia? It has to be clarified.
- In Table 3, foot note on driver mutations profile need to be described more in detail.
Author Response
In this study, authors evaluated covariates other than those commonly related to PMF which can better define prognosis in pre-PMF patients in the real-world setting. Their findings underlining the importance of other additional risk factors, such as diabetes and second neoplasia together with IPSS and DIPSS, can fill the gap to better define prognosis in pre-PMF patients. However, there are major concerns that should be addressed by the authors.
- In this study, an association was observed between IPSS and DIPSS and occurrence of death, whereas other analyzed endpoints were not associated with IPSS or DIPSS. Causes of death need to be summarized. Did events of death include only disease-related death or death which was both related and not related to disease?
We thank the reviewer for his/her comments. We have now added all the information he/she required in Table 1. Additionally, death outcome included all causes of death (disease related and non-disease related). We decided not to separate death events into the two different classes (disease related and non-disease related) due to the low accuracy of the non-disease related class.
- For second neoplasia, if possible, the information what kind of neoplasia was occurred may be useful. Did you include secondary hematologic malignancies as second neoplasia? It has to be clarified.
We have now added all the information he/she required in Table 1.
- In Table 3, foot note on driver mutations profile need to be described more in detail.
We have now added all the information he/she required in Table 3.
Reviewer 3 Report
Important and interesting work.
No remarks.
Author Response
We thank the reviewer for his/her positive comments
Round 2
Reviewer 1 Report
The authors have satisfactorily addressed all my concerns and the manuscript has improved.Therefore, I would recommend this revised version of the manuscript for publication in Cancers
Reviewer 2 Report
The authors have addressed all my comments for this manuscript.